# Identification of miRNAs and Target Genes at Key Stages of Sexual Differentiation in Androdioecious *Osmanthus fragrans*

**DOI:** 10.3390/ijms231810386

**Published:** 2022-09-08

**Authors:** Cheng Zhang, Kailu Zhang, Zihan Chai, Yanfeng Song, Xianrong Wang, Yifan Duan, Min Zhang

**Affiliations:** Co-Innovation Center for Sustainable Forestry in Southern China, College of Biology and the Environment, International Cultivar Registration Center for Osmanthus, Nanjing Forestry University, Nanjing 210037, China

**Keywords:** androdioecy, *Osmanthus fragrans*, miRNAs, gynoecium, sex differentiation

## Abstract

Androdioecy is the crucial transition state in the evolutionary direction of hermaphroditism to dioecy, however, the molecular mechanisms underlying the formation of this sex system remain unclear. While popular in China for its ornamental and cultural value, *Osmanthus fragrans* has an extremely rare androdioecy breeding system, meaning that there are both male and hermaphroditic plants in a population. To unravel the mechanisms underlying the formation of androdioecy, we performed small RNA sequencing studies on male and hermaphroditic *O. fragrans*. A total of 334 miRNAs were identified, of which 59 were differentially expressed. Functional categorization revealed that the target genes of differentially expressed miRNAs were mainly involved in the biological processes of reproductive development and the hormone signal transduction pathway. We speculated that the miRNA160, miRNA167, miRNA393 and miRNA396 families may influence the sex differentiation in *O. fragrans*. Overall, our study is the first exploration of miRNAs in the growth and development process of *O. fragrans*, and is also the first study of androdioecious plants from the miRNA sequencing perspective. The analysis of miRNAs and target genes that may be involved in the sex differentiation process lay a foundation for the ultimate discovery of the androdioecious molecular mechanism in *O. fragrans*.

## 1. Introduction

Angiosperms are also known as flowering plants and are divided into different breeding systems depending on the sex of the flowers. In all breeding systems, androdioecy is extremely rare, and there are no more than 50 species of plants recognized as androdioecy breeding systems in the world, accounting for no more than 0.05% [1]. Simultaneously, androdioecy is the crucial transition state in the evolutionary direction from hermaphroditism to dioecy or from gynodioecy to dioecy [2,3,4]. However, Oleaceae plants seem to have a special preference for such a rare breeding system, and it is almost miraculously concentrated in Oleaceae and *Osmanthus* plants, including *Phillyrea angustifolia* [5,6], *Fraxinus ornus* [7], *Chionanthus retusus* [8], *Osmanthus delavayi* [9] and other plants. *Osmanthus fragrans* belongs to *Osmanthus* of Oleaceae, widely cultivated and deeply loved in China due to its ornamental and cultural value. Next-generation sequencing (NGS) has been used to study the genome and transcriptome of *O. fragrans*, mainly concentrated in the research of flower color and fragrance [10,11,12]. Meanwhile, *O. fragrans* has both male and hermaphrodite plants, and previous studies have shown that they are mainly different in the development process of the pistil, that is, the two carpels of hermaphrodite *O. fragrans* can heal and develop into a complete pistil, but the two carpels of male *O. fragrans* remain separate, which eventually leads to pistil abortion. The mechanisms leading to the sex differentiation of this androdioecy breeding system are still unclear [13]. In addition, there are hermaphroditic and dioecious plants in the Oleaceae, thus, *O. fragrans* is also an ideal material for studying the evolution of plant reproductive systems.

MicroRNAs (miRNAs) are a class of non-coding endogenous small molecule single-stranded RNA in eukaryotes. Mature miRNAs regulate genes at the post-transcriptional level by cutting mRNA or inhibiting mRNA translation [14,15,16]. These miRNAs play a pivotal role in promoting plant growth and development [17,18,19], increasing crop yield [20], boosting plant immunity [21,22] and enhancing plant resistance [23,24]. Recent studies have shown that miRNA can be transported over a long distance in plants through the vascular system, and can also act as a communication signal molecule, moving between plants and interacting organisms (including pathogens) to induce gene silencing. This phenomenon is known as cross-kingdom/organism RNA interference (RNAi) [25,26]. Numerous studies suggest that miRNAs also play a key role in the sex differentiation of plants, with differentially expressed miRNAs regulating flower development by affecting auxin and other signaling pathways, leading to abortion of pistils or stamens, eventually evolving into sex differentiation [27]. The trioecious species papaya (*Carica papaya*) has three genders: male, female and hermaphrodite flowers. A total of 12 differentially expressed miRNAs were identified in flowers of the three genders. The highly expressed miRNAs in male flowers were involved in auxin signaling, whereas the highly expressed miRNAs in female flowers were potential regulators of apical meristem [28]. In hermaphroditic cotton (*Gossypium hirsutum*), overexpressed miR157 inhibited auxin signaling and increased the resistance to high temperature, but at the same time causes degeneration of microspores and anther indehiscence, resulting in male sterility [29]. *Ginkgo biloba*, known as a living fossil, is a dioecious plant. During the development of male and female strobili, 61 and 36 miRNAs were continuously and highly expressed, respectively. These differentially expressed miRNAs may affect the sex determination of *Ginkgo biloba* by regulating IAA and other hormone pathways [30].

We previously attempted to study the breeding system of *O. fragrans* from the perspective of sequential windowed acquisition of all theoretical mass spectra (SWATH-MS) quantitative proteomics, and the results showed that differentially expressed proteins in glucose metabolism, secondary metabolism and calcium signaling pathways may affect the sex differentiation [31]. As is known to all, important information about regulating gene function is hidden in these miRNAs, and the study of miRNAs in the genus *Osmanthus* and even the Oleaceae has not been reported. Thus, in this study, miRNA sequencing was performed in flower buds and floral tissues of androdioecious *O. fragrans* at three key developmental stages, hoping to explore the role of miRNAs in the process of sex differentiation and provide clues for ultimate mechanism research.

## 2. Results

### 2.1. Morphological Observation of Three Developmental Stages

The flower buds and flowers of male and hermaphrodite *O. fragrans* were collected every 7 days. By comparing the results of paraffin sections at different stages, we finally selected three critical periods, namely, the flower bud materials before and after sex differentiation, and the flower tissue at the full flowering stage that completed the differentiation.

In the first stage, male and hermaphrodite flower buds were in the stamen differentiation stage, and showed no difference in morphology, with two carpels and two stamens developed from one flower primordia (Figure 1M1,H1). The second stage was the pistil differentiation stage, the two carpels of male flower buds developed slowly and remained separated, while in hermaphrodite flower buds, the two carpels fused at the top and formed a complete, normal pistil (Figure 1M2,H2). The third stage was the flowering stage of male and hermaphrodite flowers, the two carpels of the male remained separate and they formed the pistil without structures such as stigma, style and ovary (Figure 1M3,M4). In contrast, after the healing of the two carpels, the hermaphrodite pistil formed and developed normally and could be fertilized to grow into fruit (Figure 1H3,H4).

### 2.2. Sequencing Quality of sRNAs and miRNA Identification

According to the section results of three stages, three biological replicates were selected for each stage of male and hermaphroditic *O. fragrans* for miRNA sequencing library construction. The BGISEQ-500 platform was used for small RNA sequencing, and a total of 292,930,383 sequencing tags were obtained, and an average of 24,410,865 tags were obtained for each sample. After filtering, the tags with sequencing quality higher than Q20 of each sample were more than 99% (Table 1). The length of sRNAs was within the range of 18–30 nucleotides, mainly distributed in 21–24 nucleotides, of which 24 nt was the most common, accounting for 57.56%, followed by 21 nt, accounting for 13.56% (Appendix A).

Depending on the comparison results of the miRBase 22.1 database and miRDeep2 prediction results, a total of 334 miRNAs were obtained, including 205 known miRNAs and 129 newly predicted novel miRNAs. They were distributed on 23 chromosomes of *O. fragrans*. Chr04 was the most numerous with 29 miRNAs, including 14 known miRNAs and 15 novel miRNAs. Chr15, Chr19 and Chr22 were the least common, with only five miRNAs (Figure 2A; Appendix A). MiRNAs were typically conserved at the sequence level, with the 205 known miRNAs belonging to 40 families, among which miR156 was the most abundant with 24 miRNAs, followed by miR166 with 16 miRNAs, and there were 16 families with only one miRNA identified (Figure 2B). Nucleotide bias analysis of all miRNAs at different positions showed that the nucleotide distribution of each position of miRNAs was relatively uniform (Appendix A), but the first nucleotide of conserved miRNAs was more biased to uracil (U), close to 60%, and the 24th nucleotide was always cytosine (C), 100% (Appendix A). The first nucleotide at the 5′ end of miRNAs of different lengths had a stronger preference for either uracil (U) or adenine (A), for example, the first nucleotide of the 23nt conserved miRNAs was always A, and the first nucleotide of the 24nt conserved miRNAs was always U (Appendix A).

### 2.3. Differentially Expressed miRNAs of Three Stages

Differentially expressed miRNAs in the same developmental period between males and hermaphrodites were identified using ratio >2 or <0.5 and *P*adj < 0.05. In the first flower buds’ developmental stage, 19 differentially expressed miRNAs were identified, of which 13 were up-regulated in male plants and six were up-regulated in hermaphrodites. In the second period, 37 differentially expressed miRNAs were identified, of which 19 were up-regulated in males and 18 were up-regulated in hermaphrodites. A total of 18 differentially expressed miRNAs were identified at the full flowering stage, of which six were up-regulated in males and 12 were up-regulated in hermaphrodites. There were four differentially expressed miRNAs shared by three development stages, all of which were newly predicted miRNAs. Among them, three miRNAs were continuously up-regulated in males, which were of-novel_mir11, of-novel_mir98 and of-novel_mir125. There was one miRNA continuously down-regulated in males, which was of-novel_mir47 (Figure 3A). The heat maps of differentially expressed miRNAs in the three stages are shown in Figure 3C,D.

### 2.4. Prediction and Annotation of Target Genes

Target genes of differentially expressed miRNAs at different developmental stages were predicted by TargetFinder and psRobot, respectively. Summarizing the results of the two, 2058 target genes of 250 miRNAs were predicted. Except for 10 miRNAs such as of-miR167a, which predicted only one target gene, the rest of the miRNAs all had multiple target genes, and of-miR5021 had the most target genes with 245. Among the 2058 predicted target genes, 1324 genes were regulated by a single miRNA. The same gene can also be regulated by multiple miRNAs, and 40 target genes were regulated by more than 10 miRNAs, for example, ofr.gene21448 had 20 corresponding miRNAs (Appendix A).

MiRNAs regulate plant growth and development by regulating the expression of target genes, so figuring out the function of target genes through GO and KEGG analysis can enable us to understand the regulatory role of the corresponding miRNAs. GO enrichment analysis was performed on target genes with differential expression of miRNAs at each stage, and the results showed that: in the first stage, the 112 target genes of 19 differentially expressed miRNAs were mainly involved in biological processes such as phyllome development, shoot system development and plant organ development, and mainly encoded auxin response factors and transcription factor TCP4-like proteins, involving ofr.gene48854, ofr.gene38648, ofr.gene54699, ofr.gene54712, ofr.gene57645 and other genes. The 335 target genes of the 37 differentially expressed miRNAs in the second stage were mainly involved in meristem initiation, determination of bilateral symmetry, specification of axis polarity and other biological processes, and mainly encoded homeobox-leucine zipper proteins (HB), involving ofr.gene24381, ofr.gene43925, ofr.gene1302, ofr.gene10587, ofr.gene56924, ofr.gene55621, etc. In the third stage, 112 target genes with 18 differentially expressed miRNAs were mainly involved in the RNA metabolic process and nucleic acid metabolic process (Appendix A). The results of KEGG pathway analysis showed that the target genes of differentially expressed miRNAs in the first developmental stage involved eight pathways, and the second and third stages involved seven and six pathways, respectively. The pathways involved in all three stages were plant hormone signal transduction, signal transduction, environmental information processing and transcription factors (Figure 4), indicating that these four pathways may play an important role in the sexual differentiation of *O. fragrans*.

### 2.5. Validation of the Regulation of Target Genes by miRNAs

Three biological replicates of male and hermaphroditic *O. fragrans* at each stage were selected for quantitative real-time PCR (qRT-PCR) to verify the expression levels of miRNAs and predicted target genes, and to determine the regulatory relationship between miRNAs and target genes. According to the prediction results, ofr.gene43925 was the target gene of of-miR166 (Appendix A), and of-miR1666 (Figure 5E) was up-regulated in the second stage of the males, meanwhile, ofr.gene43925 (Figure 5A) was down-regulated in expression in the second stage in males. Similarly, of-miR396b (Figure 5F) suppressed the expression of ofr.gene17013 (Figure 5B) in the second stage in male *O. fragrans*. Both ofr.gene11963 and ofr.gene38443 were the target genes of of-miR167k, and of-miR167k (Figure 5M) was down-regulated in the third stage of the males, meanwhile, both ofr.gene11963 (Figure 5I) and ofr.gene38443 (Figure 5J) were up-regulated in expression in the third stage in males. Ofr.gene57509 (Figure 5K) was predicted to be the target gene of the miR393 family, and its expression in the third stage of males was inhibited by up-regulated miR393h (Figure 5N) and miR393-5p (Figure 5O). However, the expression of of-miR157d with its target gene ofr.gene51165 (Figure 5L) did not conform to the regulation pattern between miRNAs and target genes, and we speculated that other mechanisms regulated the expression of ofr.gene51165.

In addition, the quantitative results of most miRNAs were consistent with the sequencing results, such as that of-miR166 (Figure 5E) was up-regulated in the second developmental stage of hermaphrodites and of-novel_mir11 (Figure 5Q) was up-regulated in males at all three developmental stages (Figure 5). In general, most of the miRNAs in this study, such as miR166, miR167 and miR393 families, had negative regulatory effects on the target genes and inhibited the expression of the corresponding target genes.

### 2.6. Verification of miRNA Target Genes by 5′RLM-RACE

To investigate the potential target genes of miRNAs and the way to regulate them, RNA ligase-mediated rapid amplification of cDNA ends (5′RLM-RACE) was used to validate two miRNAs in this study (Figure 6A), and the target genes in the corresponding period were quantitatively verified by qRT-PCR (Figure 6B).

The results confirmed that ofr.gene32426 and ofr.gene40945 were both target genes of of-miR160a-5p, and the cleavage site was between bases 10/11 of the complementary pairing of miRNAs and target genes. Quantitative results indicate that the two target genes were less expressed in males during the third stage of development, while of-miR160a-5p was up-regulated in the first two stages of males, and we speculated that of-miR160a-5p may function mainly during the third stage. The splicing site of of-miR166 on ofr.gene45828 was also between the 10th/11th bases, and in the second stage, the expression level of ofr.gene45828 in males was higher than that in hermaphrodites, which was consistent with the regulation pattern of miRNAs. Thus, the above experiments validated that miRNAs can directly inhibit the expression of their target genes by complementary pairing with them.

## 3. Discussion

The miRNAs regulate the expression of downstream target genes mainly through the interaction with transcription factors [32], and an increasing number of studies have found that some miRNAs are closely associated with sex determination in plants [33]. MiRNAs are conserved, and the function of their target genes is also relatively conservative, which has a very important inspiration for us. In dioecious persimmon (*Diospyros lotus*), the MeGI gene on the autosome inhibited stamen development to form female flowers, while the OGI gene on the Y chromosome produced small RNA to suppress MeGI gene expression to form female flowers [34]. The miRNA studies of the dioecious plant *Populus tomentosa* showed that 61 and 11 miRNAs were specifically expressed in male and female plants, respectively, and miRNA Pto-F70 and its four target genes were located in the sex-determining region of chromosome XIX [35]. The miRNA study of the dioecious asparagus (*Asparagus officinalis*) found that 37 miRNAs were highly expressed in females and 26 in males, and the predicted target genes of differentially expressed miRNAs associated with the development of floral organs may lead to sex differentiation [36]. EgmiR159a was a major factor in female flower determination in the main oil-producing crop, oil palm (*Elaeis guineesis*). High expression of EgmiR159a can increase the number of female flowers and thus increase oil production [37]. In monoecious *Xanthoceras sorbifolium*, 112 differentially expressed miRNAs were identified in males and females, 17 of which were directly involved in the development of flower and gametophyte, among which the miR393 and miR396 family may be involved in the development of stamen and ovule [38].

In this study, we determined three key developmental processes of pistils through the morphological comparison of male and hermaphrodite *O. fragrans*. Among them, the most significant difference lay in the stage of pistil differentiation. The two carpels in males were stunted and remained separated all the time, whereas the two carpels of the hermaphrodites fused at the top and eventually developed into a complete pistil (Figure 1). The number of differentially expressed miRNAs in this stage is the largest of the three stages, accounting for half of all differentially expressed miRNAs (Figure 3).

In GO enrichment analysis of predicted target genes, most of the target genes in the three stages were classified into the biological process of plant organ development, especially related to reproductive development (Appendix A). For example, the target genes encoding homeobox-leucine zipper protein (HB), regulated by the miR166 family, may affect the healing of carpels or ovule development by regulating vascular cell differentiation and the initiation of meristematic tissue, thus leading to the pistil abortion [39]. The target genes of probable indole-3-pyruvate monooxygenase (YUC) involved in auxin synthesis were regulated by the miR396 family, modulated gynoecium development or formation by influencing auxin homeostasis [40,41,42]. In addition, miR396 can also control the carpel number, pistil development and fruit development, overexpression of miR396 in Arabidopsis led to abnormal pistil development and down-regulated miR396 expression in tomato resulted in significant enlargement of flowers, sepals and fruits [43,44]. The stromal processing peptidase (SPP) encoded by ofr.gene18985 target genes regulated by the miR172 family may also be associated with gynoecium abortion. In Arabidopsis, SPP is involved in embryonic development, and its mutants lead to the abnormal development of some seeds [45]. MiR172 can also act as a translational repressor of the APETALA2 (AP2) gene, thus affecting the fate of Arabidopsis floral organs [46].

Among the results of the KEGG pathway analysis, the most significant enrichment pathway in all three stages was hormone signal transduction, and mainly associated with the auxin signaling pathway. In this pathway, almost half of the target genes were directly or indirectly involved in the response of auxin signal, namely, auxin response factors (ARFs), auxin signaling F-box proteins (AFBs) and transport inhibitor response proteins (TIRs) regulated by miR160, miR167 and miR393 families, respectively. MiR393-regulated CsTIR1 and CsAFB2 act as auxin receptors and play essential roles in auxin-mediated fruit development, and their overexpression in cucumber exhibited decreased seed germination potential and higher parthenocarpic fruit set rates [47]. The mir156 family mainly regulated the transition from vegetative growth to reproductive growth, and transcription factor squamosa promoter-binding-like proteins (SPLs) regulated by the miR156 family were involved in the gynoecium development [48,49,50]. Numerous studies had revealed that the auxin signaling pathway played a significant role in the development of the gynoecium [51,52]. In Arabidopsis, the mutation of ARF3 showed defects in tissue patterning along the longitudinal axis of the gynoecium [53,54]. Overexpression of Arabidopsis miR167a in tomatoes can target and regulate the down-regulated expression of ARF6 and ARF8, resulting in defective flower development and female sterility [55]. In the present study, auxin response factors ofr.gene40945 and ofr.gene32426 were indeed the target genes of of-miR160a-5p verified by 5′RLM-RACE experiments to be down-regulated in males (Figure 6).

Plant sex determination is an extremely complex developmental biological process, which is affected by various aspects such as the environment and genes. Studies have shown that epigenetic mechanisms may play a key role in phenotypic plasticity and rapid adaptation of plants to environmental changes [56]. As an important mechanism of epigenetic regulation, miRNAs also play an important role in the sex differentiation process in plants. We speculate that compared with sex chromosome differentiation, *O. fragrans* used miRNAs in a more rapid, flexible and direct way (Table 2 and Appendix A), which affected the gender differentiation and helped to adjust its reproductive strategy, to adapt to the changes in the environment and climate in a special period and, thus, formed androdioecy, a transitional breeding system between hermaphroditic and dioecious [57,58].

## 4. Materials and Methods

### 4.1. Sample Processing and Sequencing of Small RNA

In this study, we selected two cultivars of *O. fragrans* Albus group on the campus of Nanjing Forestry University for comparison. The whole flower buds of the stamen differentiation stage and pistil differentiation stage and the whole flower tissue with bracts of the full flowering stage were collected from May to October in 2020. Some flower buds were fixed with FAA (5 mL formalin: 5 mL acetic acid: 90 mL 70% ethanol) for paraffin sections [9,13]. Both flower buds and the whole flowers were frozen with liquid nitrogen for subsequent sequencing and validation experiments according to the results of the section.

Total RNA was extracted with TRIzol, and polyacrylamide gel electrophoresis (PAGE) was used to separate RNA into different fragments, and the fragments from 18–30 nt were collected to recover small RNA. The 5′ and 3′ ends, respectively, were connected to the adaptors to perform RT-PCR reactions and purify the PCR products [59]. Finally, small RNA sequencing was performed using the BGISEQ-500 sequencing platform. For the small RNA sequencing data, tags with low sequencing quality, 5′ end connector contamination and tags without a 3′ end connector sequence were removed. Tags that contained inserts and poly A were filtered out. Finally, the clean tag of each sample was obtained.

### 4.2. Identification of miRNAs and Differentially Expressed miRNAs

The clean tags obtained from each sample were aligned to NCBI and Rfam databases to remove the non-target RNAs (rRNAs, tRNAs, snRNAs and snoRNAs). According to the conservation of plant miRNAs, the remaining tags were compared to the miRBase 22.1 [60], and compared with the known plant miRNA precursors or mature sequences to identify the mature miRNAs. Novel miRNAs were predicted in the remaining reads using miRDeep2 [61], using the genome of *O. fragrans* as the reference sequence. DESeq2 was used to identify miRNAs that were differentially expressed (ratio >2 or <0.5 and Padj < 0.05) in male and hermaphrodite *O. fragrans* at the same developmental stage [62].

### 4.3. Prediction and Functional Annotation of Target Genes

psRobot and TargetFinder were used to predict target genes of all miRNAs, and the union of all prediction results was taken as the final prediction result [63,64]. TBtools was used for Gene Ontology (GO) enrichment and Kyoto Encyclopedia of Genes and Genomes (KEGG) pathway analysis of differentially expressed miRNA target genes, revealing the main biological processes that target genes are involved in, and exploring potential gender regulatory pathways [65].

### 4.4. Verification of miRNA Target Genes

A FirstChoice^®^ RLM-RACE Kit (AM1700 Invitrogen, Waltham, MA, USA) was used to verify the loci of miRNA target genes. The miRNAs and the predicted target genes to be verified were selected, and the corresponding primers were designed (Appendix A). An adapter was added to the RNA 5′ end, and reverse transcription was carried out. The reverse transcription products were used as templates for nested PCR and the PCR products were recovered and ligated to the pMD19-T vector (6013 Takara). The splicing sites of miRNAs were analyzed by sequencing and comparison with target gene sequences.

### 4.5. The qRT-PCR Validation of the miRNAs and Target Genes

The poly (A) polymerase was used to add a tail to the 3′ end of the miRNAs, and then universal reverse transcription primers (oligo dT primers with specific labels) were used for reverse transcription reaction to generate the first strand cDNA of the miRNAs (638315 Takara). The SYBR Green Premix Pro Taq HS qPCR Kit II (AG11702, ACCURATE BIOTECHNOLOGY, HUNAN, Co., Ltd., Changsha, China) was used for miRNA quantification according to the instructions. For target gene qRT-PCR quantification, RNAs of three biological replicates for each stage of male and hermaphroditic *O. fragrans* were reversed transcribed by an Evo M-MLV RT Kit with gDNA Clean for qPCR II (AG11711, ACCURATE BIOTECHNOLOGY, HUNAN, Co., Ltd., Changsha, China) for cDNA synthesis. A SYBR Green Premix Pro Taq HS qPCR Kit (AG11701, ACCURATE BIOTECHNOLOGY, HUNAN, Co., Ltd., Changsha, China) was used for quantitative real-time PCR (qRT-PCR) experiments with OfACT as a reference, and all primers are listed in Appendix A.

## 5. Conclusions

*O. fragrans* can be divided into male and hermaphrodite plants according to sex, and has a very rare androdioecy breeding system in nature. According to the morphological comparison between male and hermaphroditic *O. fragrans*, the main morphological differences in the development of the gynoecium between them were confirmed, that is, in the carpel development process, the two carpels of males remained separated, while the two carpels of hermaphrodites healed to form a fully developed pistil. Based on this, a small RNA sequencing study was conducted for the three key developmental stages to investigate the sex differentiation of *Osmanthus* from the perspective of miRNAs. A total of 59 differentially expressed miRNAs were found at the three stages, and were mainly concentrated in the second stage. Altogether, 2058 target genes were predicted for 250 miRNAs using psRobot and TargetFinder. The analysis of GO and KEGG showed that the target genes were involved in the biological process of reproductive development and the hormone signal transduction pathway, respectively. Target genes involved in auxin signal response were enriched in both GO and KEGG enrichment results. For example, of-miR160a-5p was up-regulated in males and the results of 5′RLM-RACE confirmed that its two target genes encoding ARF were down-regulated, and their cleavage sites were located at the 10th/11th base, consistent with the predictions. Overall, our study of miRNAs at key developmental stages undoubtedly helps us to approach the truth about the regulatory mechanisms of sexual differentiation in androdioecious *O. fragrans*. In addition, the identification of miRNAs in this study will also provide data support for other studies of *O. fragrans*.

## Figures and Tables

**Figure 1 ijms-23-10386-f001:**
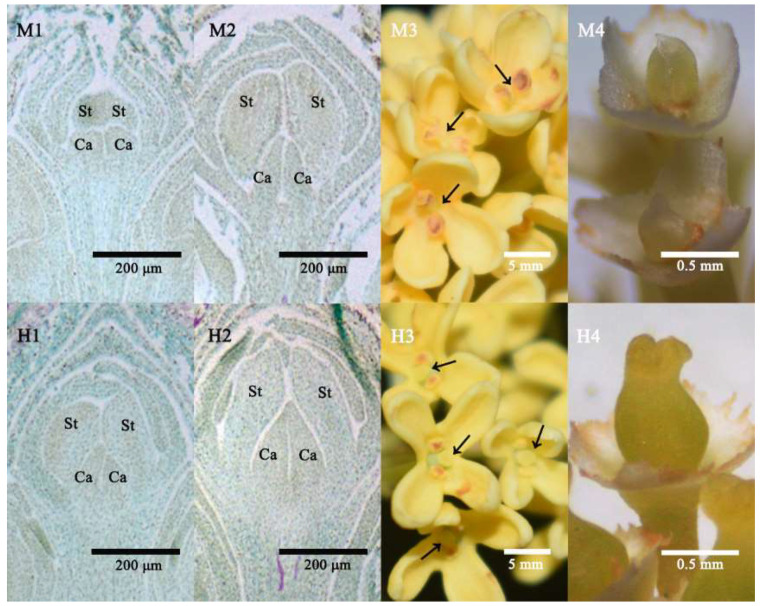
Three key developmental stages of male and hermaphrodite *O. fragrans*. (**M1**,**M2**,**H1**,**H2**) The first two stages were paraffin sections of flower buds. (**M3**,**H3**) A morphologic photograph of the third, full flowering stage, arrows indicate the position of the pistils. (**M4**,**H4**) Photographs of pistils of male and hermaphrodite flowers with petals and stamens removed. St, Stamen; Ca, Carpel.

**Figure 2 ijms-23-10386-f002:**
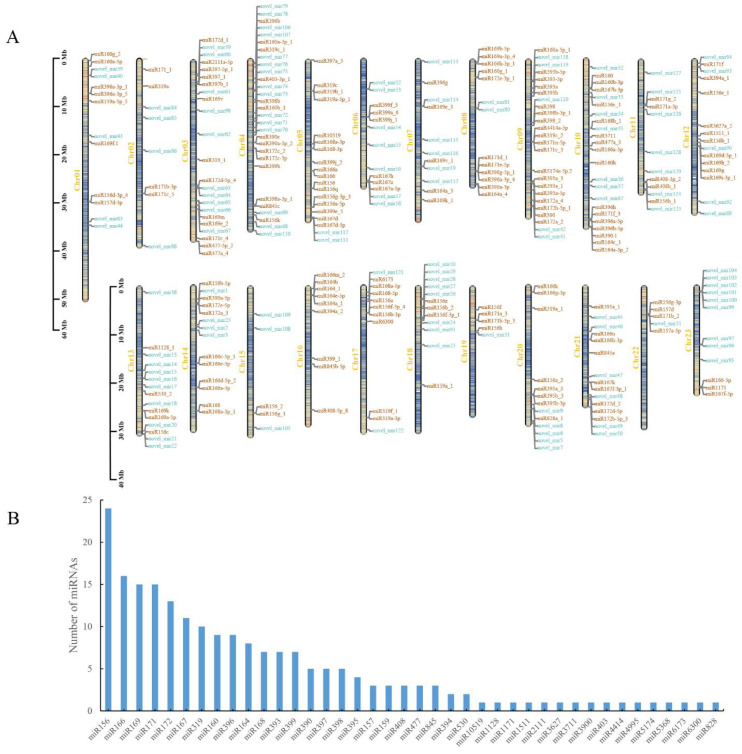
Localization of miRNAs on chromosomes and family analysis of conserved miRNAs. (**A**) Localization of identified miRNAs on the chromosomes. Orange and blue fonts indicate the known mature miRNAs and novel miRNAs, respectively. (**B**) The 40 families of conserved miRNAs.

**Figure 3 ijms-23-10386-f003:**
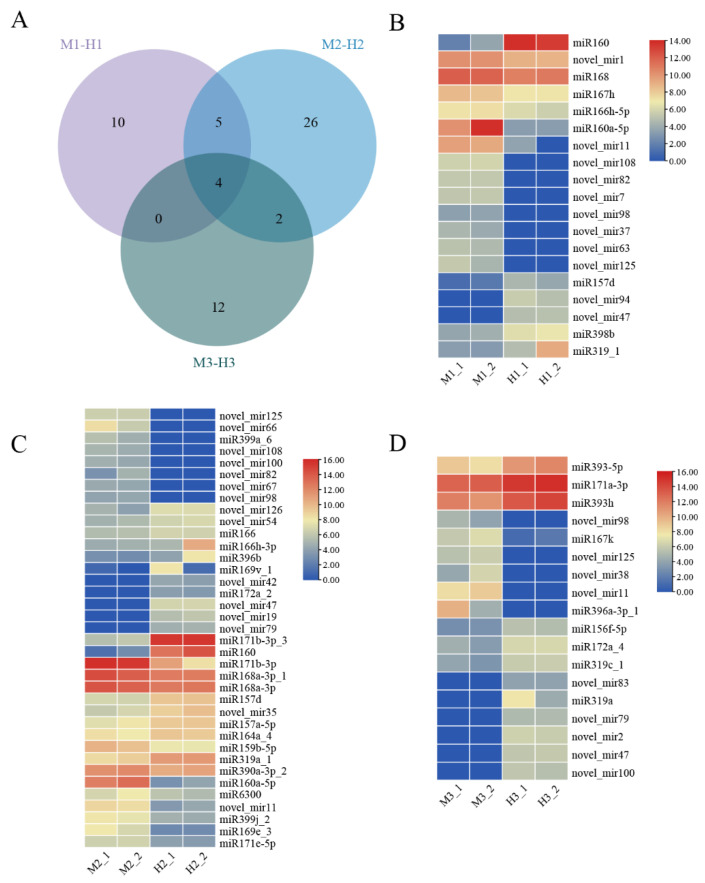
Venn diagram and heat maps of miRNAs differentially expressed in males and hermaphrodites at three stages. (**A**) Venn diagram of differentially expressed miRNAs at three stages. (**B**) Heat map of differentially expressed miRNAs in the first stage (M1-H1). (**C**) Heat map of differentially expressed miRNAs in the second stage (M2-H2). (**D**) Heat map of differentially expressed miRNAs in the third stage (M3-H3).

**Figure 4 ijms-23-10386-f004:**
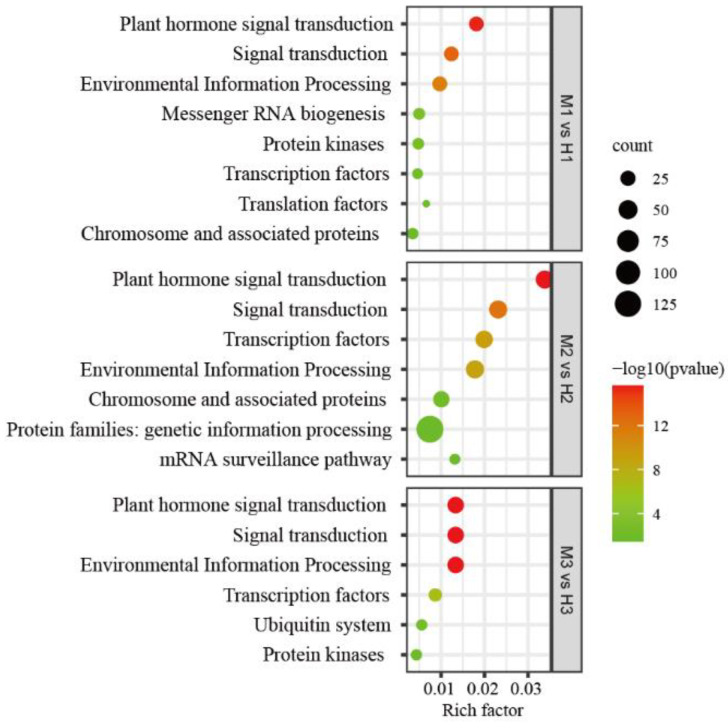
KEGG pathway analysis of target genes predicted by differentially expressed miRNAs in three stages. M1 vs. H1, M2 vs. H2, M3 vs. H3 indicate the first, second and third stage, respectively.

**Figure 5 ijms-23-10386-f005:**
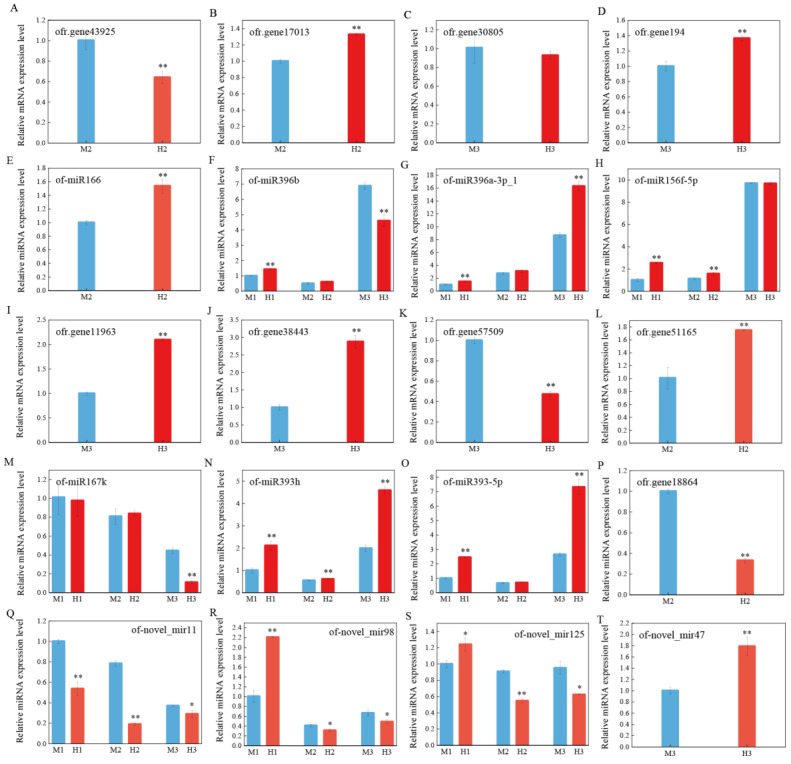
qRT-PCR validation of miRNAs and target gene expression levels at different stages. Asterisks indicate significant differences, as determined by Student’s *t*-test (** *p* < 0.01 or * *p* < 0.05). (**A**–**T**) represents the relative expression levels of mRNA or miRNA at different stages, respectively.

**Figure 6 ijms-23-10386-f006:**
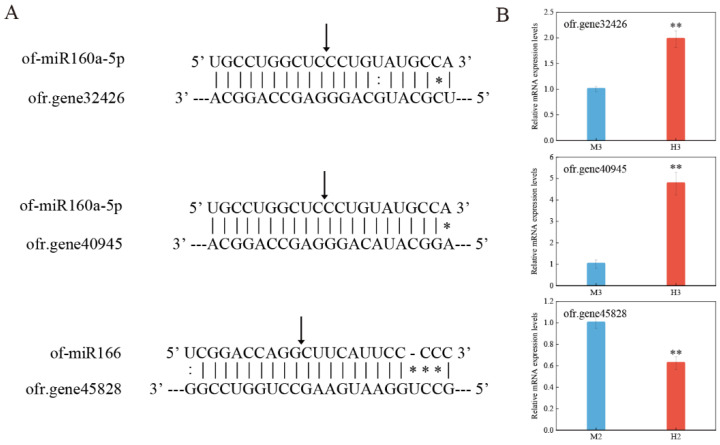
5′RLM-RACE experiment of miRNA cleavage sites and quantitative verification of target genes. (**A**) Schematic diagram of target gene cleavage sites by miRNAs. The arrows indicate the detected cleavage sites, * indicates mismatches and : indicates G–U mismatches. (**B**) Relative expression levels of target genes. Asterisks indicate significant differences, as determined by Student’s *t*-test (** *p* < 0.01).

**Table 1 ijms-23-10386-t001:** Data information for small RNA sequencing.

Sample Name	Raw Tag Count	Clean Tag Count	Q20 of Clean Tags (%)	Percentage of Clean Tags (%)
H1_1	25157374	23301337	99.1	92.62
H1_2	24620128	22686185	99.3	92.14
H2_1	24802002	23559721	99.3	94.99
H2_2	24766857	23581378	99.3	95.21
H3_1	24524362	23373531	99.2	95.31
H3_2	24387588	23001025	99.3	94.31
M1_1	24446036	22915022	99.1	93.74
M1_2	23457776	21759535	99.2	92.76
M2_1	24185224	22517684	99.4	93.11
M2_2	23626349	20571450	99.3	87.07
M3_1	24316866	22868226	99.4	94.04
M3_2	24639821	22900886	99.3	92.94

**Table 2 ijms-23-10386-t002:** The main differentially expressed miRNAs in three stages.

MiRNAs	Log2 Fold Change	DEM	Stages
of-miR156f-5p	−2.61165	Down	M3-H3
of-miR160	−10.6799	Down	M1-H1
of-miR160	−11.7911	Down	M2-H2
of-miR160a-5p	9.775465	Up	M1-H1
of-miR160a-5p	9.31752	Up	M2-H2
of-miR166	−1.19515	Down	M2-H2
of-miR166h-3p	−5.0222	Down	M2-H2
of-miR166h-5p	1.376418	Up	M1-H1
of-miR167h	1.417542	Up	M1-H1
of-miR167k	5.710173	Up	M3-H3
of-miR393-5p	−2.7161	Down	M3-H3
of-miR393h	−2.42835	Down	M3-H3
of-miR396a-3p_1	11.27262	Up	M3-H3
of-miR396b	−4.47851	Down	M2-H2

## Data Availability

The raw data of sRNA sequencing in this study are available in the National Center for Biotechnology Information (NCBI) Sequence Read Archive (SRA) database under BioProject ID PRJNA834730 (accession number SRP373346).

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
