# Peer review of "Identification of miRNAs and Target Genes at Key Stages of Sexual Differentiation in Androdioecious Osmanthus fragrans"

_ijms, 2022, doi:10.3390/ijms231810386_

Round 1
Reviewer 1 Report
Dear authors,
I had reviewed the paper with a title, Identification of miRNAs and Target Genes at Key Stages of Sexual Differentiation in Androdioecious Osmanthus fragrans, which was submitted to the Journal, International Journal of Molecular Sciences (IJMS). I believed this work is ready for the publication on IJMS.
Sincerely,
Author Response
Dear Reviewer,
Thank you for your patient work and the affirmation of our article.
Reviewer 2 Report
Dear authors,
I believe that you have a well-prepared manuscript. It seems to me that it is worth improving the presentation of information in the figures, making it more readable. Special attention should be paid to Figure 5: dotted lines are incomprehensible. It may be more convenient if the individual panels are labeled with letters.
Author Response
Thank you for your patient work and the affirmation of our article.
Thanks for your suggestion. The expression levels of the miRNAs and their target genes were in the same box composed of dotted lines. As you said, the dotted line in Figure 5 is really confusing, thus, we removed the dotted lines, and each small figure were labeled with letters.

Reviewer 3 Report
The manuscript “Identification of miRNAs and Target Genes at Key Stages of Sexual Differentiation in Androdioecious Osmanthus fragrans” submitted by Zhang et al. has significantly focused on an extremely rare androdioecy breeding system in O. fragrans. By way of small RNA sequencing on relevant male and hermaphroditic plant tissues, the authors were able initially to describe all the sequence and chromosome locations for all the potential miRNAs, and the target genes predicted from Targetfinder and psRobot. A total of 59 miRNAs (out of 334 identified) were differentially expressed with target genes mainly involving biological reproductive developmental processes, or hormone signal transduction pathways. Consequently, they were able to show that a few miRNA families (eg miRNA160, miRNA167, miRNA393, miRNA396) may influence the sex differentiation by regulating target genes for auxin signal responses and other pathways. Furthermore, the authors presented the GO enrichment results of the target genes in 3 different stages of floral development, and the qRT-PCR was applied to verify the expression levels of relevant miRNAs and their predicted target genes. Finally, the 5’RLM-RACE method was successfully used to validate two of the miRNAs. This paper describes valuable empirical research and would be a model for plant biologists, especially those engaged in dioecious or androdioecious breeding systems when development of male and female sexual organs become an issue. Finally, some of the English expression, phraseology and sentence structure in this paper probably needs editing by a native speaker.
Author Response
Response to Reviewer Comments
Thank you for your patient work and the praise for our manuscript.
We found a native English speaker to help us polish the manuscript.
Point 1: The manuscript “Identification of miRNAs and Target Genes at Key Stages of Sexual Differentiation in Androdioecious Osmanthus fragrans” submitted by Zhang et al. has significantly focused on an extremely rare androdioecy breeding system in O. fragrans. By way of small RNA sequencing on relevant male and hermaphroditic plant tissues, the authors were able initially to describe all the sequence and chromosome locations for all the potential miRNAs, and the target genes predicted from Targetfinder and psRobot. A total of 59 miRNAs (out of 334 identified) were differentially expressed with target genes mainly involving biological reproductive developmental processes, or hormone signal transduction pathways. Consequently, they were able to show that a few miRNA families (eg miRNA160, miRNA167, miRNA393, miRNA396) may influence the sex differentiation by regulating target genes for auxin signal responses and other pathways. Furthermore, the authors presented the GO enrichment results of the target genes in 3 different stages of floral development, and the qRT-PCR was applied to verify the expression levels of relevant miRNAs and their predicted target genes. Finally, the 5’RLM-RACE method was successfully used to validate two of the miRNAs. This paper describes valuable empirical research and would be a model for plant biologists, especially those engaged in dioecious or androdioecious breeding systems when development of male and female sexual organs become an issue. Finally, some of the English expression, phraseology and sentence structure in this paper probably needs editing by a native speaker.
Response 1: Thank you for your patient work and the praise for our manuscript.
We found a native English speaker to help us polish the manuscript. The following are the modified phraseology and sentences:
Line 63: pistil → pistils
Line 105: The third period was morphologic photograph of the full flowering stage, arrows indicated the position of the pistils. → The third period was a morphologic photograph of the full flowering stage, arrows indicated the position of the pistils.
Line 151/154: heatmaps → heat maps
Line 156/157: Heatmap → Heat map
Line 195-196: Quantitative real-time PCR (qRT-PCR) was used to verify the expression levels of miRNAs and predicted target genes, so as to determine the regulatory relationship between miRNAs and target genes. → Quantitative real-time PCR (qRT-PCR) was used to verify the expression levels of miRNAs and predicted target genes, to determine the regulatory relationship between miRNAs and target genes.
Line 209-210: and we speculated that there were other mechanisms that regulated the expression of ofr.gene51165. → and we speculated that other mechanisms regulated the expression of ofr.gene51165.
Line 240: difference → differences
Line 341: clean tag → clean tags
Line 392: it → its
Reviewer 4 Report
In this research article, Zhang et al characterized the regulation of androdioecy profiling the miRNAs at three time points during Osmanthus fragrans flower development. They have identified differentially expressed miRNAs in male compared to hermaphrodite flowers and validated some misregulated miRNAs and their associated target genes. Finally, they propose that some miRNAs control auxin signaling which in turn regulates sexual differentiation in O.fragrans.
I think that the experiments were technically well performed and that the data presented is clear. The connection to auxin remains however elusive and I feel that the miRNA data presented does not suggest a role for this pathway during O. fragrans sexual differentiation. It would be important to bring some nuance and tone down this aspect of the manuscript.
Major point:
- Because auxin signaling is a complex signaling pathway, including a multi-layer regulatory structure (biosynthesis, signaling, positive and negative regulators…), hormone transport, feedback loops, domain specific input and output, I feel that the miRNA data presented is not sufficient to support a role of this hormone in that process.
The data show the regulation of 2 homologs of the ARFs by mir160, it would be important to show, alignments with other ARF members and comment on which ARFs these factors encode for and their potential function in other species. Furthermore, their KEGG pathway analysis does not show an enrichment for auxin factors.
In Fig.7 the authors draw regulatory connections that are actually not validated in their dataset but were taken from other studies. It is important to base the schematic on data that is shown in the article or clearly indicate that it comes from other studies on the schematic.
I would remove this schematic and strongly tone down the part claiming a role of auxin signaling during that process although it might be known from other species.
Minor comments:
- How much does the miRNA dataset from this study overlap with miRNA from other studies described in the Discussion from line 239-250? Are there common miRNA regulated? This could hint towards conserved key regulators.
- Fig5: y-axis legends are missing (“Relative mRNA expression level”)
Author Response
Thanks for your valuable suggestions, which are of great help to the improvement of our article.

Reviewer 5 Report
This is a well written paper that presents well-supported results that contribute to our understanding of the developmental processes associated with androdioecy. My main recommendations are to present some missing methods information such as; field identification of key developmental stages, sequencing library preparation and initial filtering, and citation of the reference genome, for completeness and repeatability.
Comments to the authors
This study investigates microRNA expression differences associated with male and hermaphrodite flowers of a plant species with a rare androdioecious mating system. The focus on microRNA expression differences is fairly novel and provides useful new insights into developmental mechanisms contributing to androdioecy. The microscopy study to identify key developmental stages provides additional insights into androdioecy development in this species and adds rigour to the later miRNA analyses. Initial differential expression results are followed up with supporting qPCR and 5'RLM-RACE experiments.
In the results/methods, some more detail is needed at some points. For example, how did the authors go from microscopic identification of key developmental stages to sampling of tissues at the desired stages? More detail about sequencing library preparation is also needed for completeness. The reference genome used to inform some of the analyses should be cited.
In the discussion, it would be interesting to learn if any links were found between the findings of this study and earlier proteomics work cited for Osmanthus fragrans.
Overall, this is a well-written paper that presents well-supported results that contribute to our understanding of the developmental processes associated with androdioecy.
Specific comments
L22-23 I would recommend adding how innovative your study might be for the wider field of androdioecy.
L33-34 I would add that transitions from gynodioecy to dioecy are also hypothesized.
L45 Start a new statement after "pistil abortion".
L66 State the breeding system of cotton.
L74 Spell out SWATH_MS acronym at first use.
L79 State the tissue(s) used.
L85-88 State how these stages can be identified without microscopy observation. In other words, how did you know to sample these stages for miRNA extraction?
L104 It would be worth starting this section with a line or two about sequencing library preparation. How many biological and technical replicates were included?
L113 Mention the source of the reference genome used for mapping to chromosomes.
L116 Replace "sequentially conserved" with "conserved at a sequence level".
L124 Replace "uracil (U) and adenine (A)" with "either uracil (U) or adenine (A)"
Table 1. Explain percentage of clean tag. Were additional filters applied after Q20?
L182 Four pathways rather than three have been listed here.
L188-190 Mention the number of replicates tested as part of these qPCR experiments.
Figure 5. It would be preferable to include error bars in this figure. Explain the dashed lines linking panels in the legend.
L214 Spell out RLM-RACE acronym at first use.
L236-239 I suggest to split this long statement.
L258 Replace "lied" with "lay"
L272. Start new statement after "development"
L319-320 More detail is needed about how these stages could be identified in the field. See comment to L85-88
L327-328 Rewrite this statement for clarity. Some more detail about the adapters is needed.
L330 Some more detail about the filtering criteria and software used are needed.
L337 Reference the genome of O. fragrans.
L338 These fold changes are different to those listed at L134
L364-367 Include the number of sample replicates used for qPCR.
L382 Specify what you are referring to by "both"
L382 Replace "it" with "its"
Author Response
Response to Reviewer Comments
Thanks for your valuable suggestions, which are of great help to the improvement of our article.
Point 1: In the results/methods, some more detail is needed at some points. For example, how did the authors go from microscopic identification of key developmental stages to sampling of tissues at the desired stages? More detail about sequencing library preparation is also needed for completeness. The reference genome used to inform some of the analyses should be cited.
Response 1: Thank you for your comments. We collected large numbers of samples at regular intervals and kept them in the refrigeratorfor standby after quick freezing in liquid nitrogen. According to the results of the section, we select three stages, and then find the samples at the corresponding stages in the refrigerator for miRNA sequencing. In other words, samples for sections and miRNA sequencing were collected at the same time.
Three biological replicates were selected for each stage of male and hermaphroditic Osmanthus fragrans for miRNA sequencing library construction. Specific library construction process is very complicated and will take a large space, so we cite a reference.
Although the genome of O. fragrans has been published [10], we sequenced and assembled a genome ourselves, which is going to write another article, and it hasn't been published yet. We hope you can understand our position.
[10] Yang, X.L.; Yue, Y.Z.; Li, H.Y.; Ding, W.J.; Chen, G.W.; Shi, T.T.; et al. The chromosome-level quality genome provides insights into the evolution of the biosynthesis genes for aroma compounds of Osmanthus fragrans. Hortic. Res. 2018, 5:72. doi: 10.1038/s41438-018-0108-0
Point 2: In the discussion, it would be interesting to learn if any links were found between the findings of this study and earlier proteomics work cited for Osmanthus fragrans.
Overall, this is a well-written paper that presents well-supported results that contribute to our understanding of the developmental processes associated with androdioecy.
Response 2: Thank you. We couldn't agree more. We have also checked whether they had some gene overlap, and we didn't find.
We previously attempted to study the breeding system of O. fragrans from the perspective of quantitative proteomics, and the results showed that differentially expressed proteins in glucose metabolism, secondary metabolism and calcium signaling pathways may affect the sex differentiation. In this study, we performed small RNA sequencing and speculated that the miRNA families may influence the sex differentiation by regulating auxin signal and other pathways.
These two studies revealed that different influencing factors may participate in the sex determination mechanism of O. fragrans from proteome and miRNA aspects, which also illustrates the complexity of the sex determination mechanism in plants.
Specific comments:
L22-23 I would recommend adding how innovative your study might be for the wider field of androdioecy.
Response 1: Thank you. We emphasized that “and is also the first study of androdioecious plants from the miRNA sequencing perspective.” (L25-26)
L33-34 I would add that transitions from gynodioecy to dioecy are also hypothesized.
Response 2: Yes, it is! We changed it to “Simultaneously, androdioecy is the crucial transition state in the evolutionary direction from hermaphroditism to dioecy or from gynodioecy to dioecy.” (L37)
L45 Start a new statement after "pistil abortion".
Response 3:Thank you. It has been corrected.
L66 State the breeding system of cotton.
Response 4: “In hermaphroditic cotton (Gossypium hirsutum)” has been changed in line 70.
L74 Spell out SWATH_MS acronym at first use.
Response 5: “SWATH-MS (Sequential Windowed Acquisition of All Teoretical Mass Spectra)” has been changed in line 78.
L79 State the tissue(s) used.
Response 6: “miRNA sequencing was performed in androdioecious O. fragrans at three key developmental stages” has been changed to “miRNA sequencing was performed in flower buds and floral tissues of androdioecious O. fragrans at three key developmental stages” (L84)
L85-88 State how these stages can be identified without microscopy observation. In other words, how did you know to sample these stages for miRNA extraction?
Response 7: We collected large numbers of samples at regular intervals and kept them in the refrigeratorfor standby after quick freezing in liquid nitrogen. According to the results of the section, we select three stages, and then find the samples at the corresponding stages in the refrigerator for miRNA sequencing. In other words, samples for sections and miRNA sequencing were collected at the same time.
L104 It would be worth starting this section with a line or two about sequencing library preparation. How many biological and technical replicates were included?
Response 8: We added this sentences “According to the section results of three stages, three biological replicates were selected for each stage of male and hermaphroditic O. fragrans for miRNA sequencing library construction.” (L113-115)
L113 Mention the source of the reference genome used for mapping to chromosomes.
Response 9: Although the genome of O. fragrans has been published [10], we sequenced and assembled a genome ourselves, which is going to write another article, and it hasn't been published yet. We hope you can understand our position.
L116 Replace "sequentially conserved" with "conserved at a sequence level".
Response 10: Thank you. It has been corrected.
L124 Replace "uracil (U) and adenine (A)" with "either uracil (U) or adenine (A)"
Response 11: Thank you. It has been corrected.
Table 1. Explain percentage of clean tag. Were additional filters applied after Q20?
Response 12: There is no additional filters applied after Q20. The filtered data are called clean tags, and Q20 represents the comparison rate between clean tags and the reference genome. The higher the Q20 (percentage), the higher the similarity between the sample and the reference genome. Percentage of clean tag(%)=Clean tag count/Raw tag count.
L182 Four pathways rather than three have been listed here.
Response 13: Thank you. It has been corrected.
L188-190 Mention the number of replicates tested as part of these qPCR experiments.
Response 14: We changed to “Three biological replicates of male and hermaphroditic O. fragrans at each stage were selected for Quantitative quantitative real-time PCR (qRT-PCR) was used to verify the expression levels of miRNAs and predicted target genes, so asand to determine the regulatory relationship between miRNAs and target genes.” (L200-201)
Figure 5. It would be preferable to include error bars in this figure. Explain the dashed lines linking panels in the legend.
Response 15: In fact, error bars were included in Figure 5. The whole figure is very big, and error bars were to small to be seen. The expression levels of the miRNAs and their target genes were in the same box composed of dashed lines. The dashed line in Figure 5 is really confusing, thus, we removed them, and each small figure were labeled with letters.
L214 Spell out RLM-RACE acronym at first use.
Response 16: We added “RNA Ligase Mediated Rapid Amplification of cDNA Ends” in line 229.
L236-239 I suggest to split this long statement.
Response 17: We deleted this sentence “which miRNAs and their target genes play a role in the gender differentiation of plants” to make it more concise. (L255-256)
L258 Replace "lied" with "lay"
Response 18: Thank you. It has been corrected.
L272. Start new statement after "development"
Response 19: Thank you. It has been corrected.
L319-320 More detail is needed about how these stages could be identified in the field. See comment to L85-88
Response 20: As we responsed before, samples for sections and miRNA sequencing were collected at the same time. Here, we changed it to “Both flower buds and the whole flowers were frozen with liquid nitrogen for subsequent sequencing and validation experiments according to the results of the section.” (L340-342)
L327-328 Rewrite this statement for clarity. Some more detail about the adapters is needed.
Response 21: Thank you for your comment. It will take a lot of space to write the process of library construction of small RNA, which is very complicated. Thus, we cite a literature here, we hope you can understand.
[59] Hafner, M.; Landgraf, P.; Ludwig, J.; Rice, A.; Ojo, T.; Lin, C.; et al. Identification of microRNAs and other small regulatory RNAs using cDNA library sequencing. Methods 2008, 44, 3-12. doi: 10.1016/j.ymeth.2007.09.009
L330 Some more detail about the filtering criteria and software used are needed.
Response 22: We added this details “For the small RNA sequencing data, tags with low sequencing quality, 5' end connector contamination and tags without 3' end connector sequence were removed. Filter out tags that contain inserts and poly A. Finally, the clean tag of each sample was obtained.” (L348-351)
L337 Reference the genome of O. fragrans.
Response 23: Although the genome of O. fragrans has been published [10], we sequenced and assembled a genome ourselves, which is going to write another article, and it hasn't been published yet. We hope you can understand our position.
L338 These fold changes are different to those listed at L134
Response 24: They are the same. “log2FoldChange > 1 or < -1” is equal to “ratio >2 or <0.5”, to avoid misunderstanding, we changed them all to “ratio >2 or <0.5” (L145)
L364-367 Include the number of sample replicates used for qPCR.
Response 25: We changed it to “For target genes qRT-PCR quantification, RNA of three biological replicates for each stage of male and hermaphroditic O. fragrans was were reversed transcribed by…” (L383-384)
L382 Specify what you are referring to by "both"
Response 26: We changed it to “Target genes involved in auxin signal response were enriched in both GO and KEGG enrichment results” (L404)
L382 Replace "it" with "its"
Response 27: Thank you! It has been corrected.